# A Determination of the Caffeine Content in Dietary Supplements According to Green Chemistry Principles

**DOI:** 10.3390/foods12132474

**Published:** 2023-06-24

**Authors:** Oktawia Kalisz, Sylwia Studzińska, Szymon Bocian

**Affiliations:** Chair of Environmental Chemistry and Bioanalytics, Faculty of Chemistry, Nicolaus Copernicus University in Toruń, 7 Gagarin St., 87-100 Toruń, Poland; 296587@stud.umk.pl (O.K.); kowalska@chem.umk.pl (S.S.)

**Keywords:** caffeine, green chemistry, liquid chromatography, dietary supplement, green chromatography, ethanol

## Abstract

Caffeine is a natural psychoactive substance that belongs to a group of chemical compounds called purine alkaloids. Caffeine is found in various plants such as coffee, tea, cocoa, guarana, and yerba mate. It is often added to dietary supplements for its ability to increase metabolism and aid in weight loss. To determine the caffeine content in dietary supplements, a novel UHPLC method was developed, compatible with the rules of green analytical chemistry. The developed method used only water and ethanol for sample preparation and chromatographic separation on a short C18 column. The obtained method confirmed that caffeine may be analyzed using only environmentally friendly solvents, ethanol, and water. The developed method is characterized by its low limit of quantitation, equal to 0.047 µg/mL, and good reproducibility (a relative standard deviation lower than 1.1%). The obtained results show that the caffeine content in tested dietary supplements is 4–35% higher than the declared amount in most cases. In comparison, the caffeine content of the drug determined using this method was performed with an accuracy of 0.4% RSD.

## 1. Introduction

Analytical chemistry, including chromatography, determines the various substances introduced into the environment and found in food, among other things. Unfortunately, analytical methods generate wastes of both organic solvents and disposable plastics, among other things. Breaking this cycle and developing methods that do not generate problematic waste is prudent. These will be methods that follow the principles of green chemistry, including green analytical chemistry [1]. An example of this may be the method for determining caffeine.

Caffeine is a natural psychoactive substance that belongs to a group of chemical compounds called purine alkaloids [2,3]. It is an organic chemical compound whose sum formula is C_8_H_10_N_4_O_2_ [4]. It is found in various plants such as coffee, tea, cocoa, guarana, and yerba mate. It is a common ingredient in many beverages, including coffee, tea, energy drinks, and sodas [4,5,6,7].

Caffeine acts as a stimulant, increasing alertness and arousal levels, improving attention span, and reducing fatigue and drowsiness. Therefore, it is commonly used as a stimulant to enhance mental and physical performance, which attracts people who want to improve their performance in sports or training [8,9,10,11,12]. Thus, it is often added to dietary supplements for its ability to increase metabolism and aid in weight loss. It can also help to reduce appetite and increase thermogenesis, increasing the body’s energy expenditure due to increased heat production. This can help with fat-burning and weight loss [2,11]. However, it is essential to remember that excessive caffeine consumption can lead to unwanted side effects, such as insomnia, irritability, anxiety, and even heart problems [4,5,8,9]. Therefore, consuming caffeine in moderation and as the dietary supplement manufacturer recommends is essential. It is also worth remembering that caffeine can affect different people differently, depending on the body’s individual sensitivity to the substance [12,13].

One of the most popular and simplest ways of determining the caffeine content in dietary supplements is by using liquid chromatography, which allows for caffeine to be separated from the other substances in a dietary supplement and directly identified. This method is very accurate and is used by research laboratories [4,14,15,16,17,18,19,20,21]. Another way is to use UV-Vis spectroscopy, which allows for the measurement of the light absorption by caffeine in a dietary supplement. This method is quick and easy to perform but is less accurate than liquid chromatography [2].

There are different methods for determining caffeine depending on the type of sample, of which high-performance liquid chromatography (HPLC) is the most widely used in the food and pharmaceutical industries [14]. Caffeine absorbs ultraviolet radiation with a maximum wavelength of about 272–273 nm, so the use of HPLC with a UV-Vis detector makes it possible to separate it from other substances and then detect it based on its characteristic absorption of radiation [2,4]. The moderately hydrophobic nature of caffeine means that it is usually separated using reverse-phase liquid chromatography HPLC, which uses a non-polar stationary phase, such as silica modified with octadecyl groups (C18) [3,22]. In contrast, the mobile phase usually consists of organic solvents (acetonitrile or methanol) with water and additives, such as acetic acid, trifluoroacetic acid, ammonium acetate, or phosphate buffer [4].

Andrews and co-workers attempted to analyze 53 dietary supplements for their caffeine content, 28 of which had information about a specific amount of caffeine on their packages. The sample preparation stage consisted of a mass uniformity test, heating the aqueous sample solution and draining it through filter paper and a 0.45 μm pore diameter filter. This was followed by an analysis using HPLC. A C18-packed column was used as the stationary phase and the mobile phase was a mixture of 0.1% H_3_PO_4_ in water and acetonitrile. Gradient elution was used, and the analysis time was 30 min. For 25 preparations with the manufacturer’s stated caffeine level, the result differed from the theoretical value by between −16% and +16%; this difference was much more significant for the remaining three products [19]. Similar results were observed by Wolonkiewicz and co-workers [19]. Regarding dietary supplements, the caffeine content ranged from 58% to 103%, while regarding drug products, the result deviated from the theoretical value by only 0.7%, confirming the method’s reliability [20]. This group also applied the reversed-phase liquid chromatography (RP LC) method using a C18 stationary phase, and the mobile phase was composed of water and acetonitrile in a volume ratio of 90:10. The analysis time was long and equal to 20 min.

On the other hand, the sample preparation method involved an easy procedure based on the extraction of aqueous solutions via ultrasound and filtration through membrane filters with a pore diameter of 0.45 μm. This method appears environmentally friendly since organic solvents are not used, but it does not allow for removing the so-called matrix effect. Additionally, the problem of caffeine solubility in pure water is commonly observed [23]. The extraction of caffeine from supplements or drugs usually requires the use of organic solvents, such as, e.g., acetonitrile [20]. Ivanova and co-workers applied a mixture of water and acetonitrile in a volume ratio of 65:35 to increase the solubility of caffeine directly from powdered tablets and capsules. The caffeine was then extracted using ultrasound and filtered twice.

The developed method was used to determine the dietary supplements’ caffeine content designed for weight reduction. They undertook an analysis of 20 products, 7 of which had a declaration of their caffeine content in the formulation. For the analysis, RP LC was applied with a C18 stationary phase and mobile phase composition of water with acetonitrile in a ratio of 65:35 (*v*/*v*). Isocratic elution was used and the analysis time was only 2 min. The caffeine contents were in moderate concentrations in almost all the samples analyzed. For preparations with a certain amount of caffeine, the values ranged from −20% to +40% [21].

Green liquid chromatography is an approach to liquid chromatography that aims to minimize the technique’s environmental impact by using more environmentally friendly solvents, media, and separation methods. The methods listed above used acetonitrile or methanol as organic modifiers, which are not green solvents [24,25,26,27]. Thus, the study aimed to develop and validate a green method for caffeine determination using ethanol as an organic modifier and a simple extraction method that allows for a high recovery and eliminates the application of toxic organic solvents.

## 2. Materials and Methods

### 2.1. Materials and Reagents

A Kromasil Ethernity C18 column (2.1 × 50 mm) with a particle size of 2.5 µm was applied. Water was prepared with a Milli-Q Water Purification System (Millipore Corporation, Bedford, MA, USA). HPLC-grade ethanol was obtained from J.T. Baker (Avantor, Radnor, PA, USA). The caffeine standard was purchased from Sigma-Aldrich (Saint Luis, MO, USA).

### 2.2. Instruments

All the experiments were conducted on the Shimadzu i-Series 2060C 3D system (Kioto, Japan). This instrument includes a quaternary solvent delivery pump with an online degasser, autosampler, column thermostat, spectrophotometric diode-array UV-Vis detector (DAD), and data acquisition station. The data were collected in LabSolutions software.

### 2.3. Samples

One medicine (R1) and six dietary supplements were tested in the study. The sample characteristics declared by the manufacturers are listed in Table 1. The declared contents of caffeine are listed in parts 3.2 and 3.3 Capsule shells were not dissolved.

### 2.4. Methods

#### 2.4.1. Uniformity of the Mass

To determine the average weight of one tablet or capsule, a weight uniformity test was carried out following the standards of the Polish Pharmacopoeia XI. It consisted of weighing 20 randomly selected tablets or capsules of each preparation on an analytical mass balance, calculating their average weights and standard deviations.

#### 2.4.2. Preparation of Sample

Four tablets (or the contents of four capsules) were ground in a mortar, weighed (20 mg each), quantitatively transferred to a 50 mL volumetric flask, and dissolved in a mixture of water and ethanol in a volume ratio of 90:10. The solutions were placed in an ultrasonic bath for 10 min. Triplicate solutions were prepared for each formulation. The solutions were diluted to a theoretical concentration of 0.03 mg/mL and then filtered through membrane filters with a pore diameter of 0.2 µm.

#### 2.4.3. Chromatographic Method

All the measurements were undertaken with the mobile phase’s 0.3 mL/min flow rate. The column thermostat was set to 30 °C, while the autosampler temperature was set to 5 °C. The injection volume was 1 µL. The detection was performed at 270 nm. If not specified otherwise, the measurements were made in triplicate. The separation was performed using a binary ethanol/water mobile phase in gradient elution.

#### 2.4.4. Quantitative Analysis of Caffeine

For the quantitative determination of caffeine, the standard addition method was chosen. The solutions for the analysis were prepared directly in chromatographic vials by mixing 1 mL of the sample extract and 0, 75, 150, and 225 µL of a caffeine standard solution with a concentration of 0.2 mg/mL, respectively.

#### 2.4.5. Method Validation

The obtained analytical method was validated to demonstrate that it was suitable for its intended purpose. The limit of detection (LOD) and quantification (LOQ) were determined experimentally based on the signal-to-noise ratio (LOD = 3 ∗ S/N and LOQ = 10 ∗ S/N).

Two caffeine solutions with concentrations of 0.1 mg/mL and 2 mg/mL (in a mixture of water and ethanol in a volume ratio of 90:10) were prepared for the linearity study. They were diluted to obtain standard solutions with the following concentrations: 6.25 ∙ 10^−5^ mg/mL, 2.5 ∙ 10^−4^ mg/mL, 0.001 mg/mL, 0.01 mg/mL, 0.1 mg/mL, and 0.25 mg/mL. The results were expressed as determination coefficients (R^2^).

Inter-day and intra-day precision studies were performed for the solutions prepared in the extract with additions of 75, 150, and 225 µL of the caffeine standard (0.2 mg/mL). For the intra-day precision, the samples were injected ten times. The samples were reanalyzed 3 and 7 days later for the inter-day precision study by injecting previously prepared samples.

#### 2.4.6. Greenness of the Method

The greenness of the method was performed according to the AGREE calculator (Gdańsk University of Technology, Gdańsk, Poland) [28]. To obtain a result from the AGREE calculator, it was necessary to download the free software available at [29] and enter all the required data. The calculator can adjust the weights for each of the 12 principles to consider (subjectively) the relevance of the introduced solutions to the segment. Automatically, the weight value for each element is set at level two, but it can be reduced to unity or increased to 3 or 4. For the developed method, it was decided to increase the weights of segments 4, 9, and 11 to distinguish the simplification of the sample preparation step, the energy efficiency of the procedure, and the use of only “green” solvents.

## 3. Results and Discussion

### 3.1. Method Development

The study’s main goal was to improve the chromatographic method for caffeine analysis according to the rules of green chemistry (a minimal use of organic solvents and a short time). For this reason, the study’s first attempt was to obtain a method that used only water as its mobile phase. The Fortis H2o column (5 µm, 150 × 4.6 mm) was tested to achieve this. Unfortunately, the retention time of the caffeine was extended and a poor peak shape was obtained. Thus, the second approach was to apply ethanol as an organic modifier of the mobile phase. Adding ethanol improved the separation; however, the idea of purely water separation was lost.

In such a case, the application of ethanol allowed for the use of typical columns for the RP LC system. As a result, the short and narrow column (2.1 × 50 mm) Kromasil Eternity was applied. Applying a short column solved the problem of a high back pressure caused by the higher viscosity of the ethanol.

Different gradient profiles were tested to improve the method parameters. The chosen gradient profile was as follows: an initial condition of 10% of EtOH with an increase to 20% over 2.5 min, then column washing at 95% of EtOH from 2.51 to 3.50, and column equilibration at the initial conditions. The total volume of ethanol per analysis was 0.6 mL. A simple sample preparation was applied, consisting only of dissolving the tablet and filtering the solution. Sample purification was not performed. This causes some of the tablet/capsule components to maybe be injected into the column. Thus, we provided a column washing step with a high-elution-strength mobile phase (95% of EtOH). This allowed for the washing of all the interfering compounds from the sample.

The obtained method allowed for a determination not only of the caffeine (peak no. 2) in the supplements but also the paracetamol (peak no. 1) extracted with caffeine from the tested medicine at this wavelength (Figure 1A). Some minor peaks were also detected, but they were not identified. In the case of most supplement extracts, only the peak of caffeine was observed at the detection wavelength of 270 nm. An exemplary chromatogram is presented in Figure 1B. Other compounds from the supplements were not dissolved or not detected by the UV detector.

### 3.2. Method Validation

The quantification method of HPLC-DAD was validated to determine the caffeine in the dietary supplements. The linearity range of the method was determined based on six different concentrations. The obtained parameters, such as the calibration curve equation, linearity range, and determination coefficient, are listed in Table 2. The determined LOD and LOQ were very low, at 0.016 and 0.047 µg/mL, respectively. These were much less than necessary for the intended application. However, the low values of the LOD and LOQ allowed for the application of the developed method to other samples. The method’s sensitivity and linearity range significantly exceeded the analytical needs of the method. In practice, the working range of the method can be considered as 0.01–0.1 mg/mL.

The method’s accuracy was tested using a sample with a certified quantity of caffeine (R1, Table 1). The developed method allowed for the caffeine content to be determined with an error lower than 0.5%. Detailed results are presented in Table 3. The RSD in Table 3 is calculated for three samples in triplicate.

The results of the repeatability (intra-day) tests are listed in Table 4. Measurements were carried out for the spiked solution of R1 with a certified caffeine content to include the effect of the matrix. The value of RSD did not exceed 0.55%. Similar results were obtained for the reproducibility test (inter-day precision). For the spiked amounts of 75 µL, 150 µL, and 225 µL, the RSDs were lower than 0.51%, 0.34%, and 0.23%, respectively.

### 3.3. Sample Analysis

The validated method was applied to determine the caffeine contents in six dietary supplements. Detailed results are listed in Table 5. The determined amounts of caffeine were higher than the declared values for all the products. Only for one dietary supplement was this difference lower than 5%. The caffeine contents were 19–36% higher than the declared amounts among the rest of the tested samples. The manufacturers of dietary supplements commonly increase the contents of the active substance in relation to the declared value in order to prevent losses resulting from a possible decomposition of the substance during storage. Such a procedure is intended to increase the chances that the content of the substance will be in accordance with the declared value when the consumer takes a dose of the preparation. This practice may explain the high caffeine contents in the analyzed preparations. The RSDs in Table 5 were calculated for three samples in triplicate.

### 3.4. Ecological Aspect of the Developed Method

In the first segment of AGREE, the mode in which the sample preparation step occurred was evaluated. The developed method’s most fitting choice was the at-line mode, since the sample preparation process was not directly connected to the analytical instrument. Still, it was close (the same room). Next, the principle of seeking to minimize the number and size of the samples was evaluated. To prepare each solution for analysis, 20 mg of each sample was weighed out, which, according to the calculator’s conversion, was within the range for semi-microanalysis (10–100 mg). Due to the sensitivity of the analytical balance, the sample weights could not be satisfactorily low from the point of view of green chemistry, for which microanalysis (1–10 mg) and ultramicroanalysis (<1 mg) are ideal. Another parameter considered was the distance of the location of the measuring device, since reducing this allowed for a reduction in the time between analyses and between taking a sample and obtaining its analytical information. The measuring apparatus was located in the same room where the whole process of the sample preparation took place, so the at-line analysis was chosen. The ideal choice in this subsection would have been an in-line analysis involving measurements occurring directly in the material under study, allowing for the sample preparation step to be bypassed altogether. The fourth segment evaluated the number of steps in the analytical procedure, which was reduced by a simplified sample preparation step. The method developed consisted of a study of the mass uniformity of the preparations and the preparations of solutions being subjected to ultrasound-assisted extraction, followed by filtration and a chromatographic analysis. The next subsection analyzed the levels of automation and miniaturization in the procedure. The developed method used UHPLC apparatus equipped with an autosampler and LabSolutions software, enabling the “Quick Batch” function, which allows for quick and intuitive sequence generation. Therefore, it was decided that the procedure could be considered semi-automatic.

On the other hand, due to apparatus limitations, it was not possible to miniaturize it; this was due to the insufficient sensitivity of the analytical balance. The sixth segment evaluated the use of the derivatization process in the analytical procedure. This process should be avoided from the point of view of green chemistry, as it involves the consumption of additional reagents, prolongs the entire procedure, and contributes to the generation of more waste. The developed method avoided this step, which is also an advantage. The seventh sub-point concerned the avoidance of waste generation and proper waste management. The analysis of one sample alone in the method used took 10 min, while the flow rate of the mobile phase was 0.3 mL/min, which meant that 3 mL of the mobile phase (including 0.6 mL of ethanol) was consumed for a single analysis. According to the calculator’s converter, this result fits the assumptions of green chemistry. Another parameter evaluated was the number of compounds analyzed during one analysis and the analysis duration. The ideal result in this subsection would be the shortest possible analysis of multiple compounds simultaneously. The developed method determined only caffeine, and despite its short retention time (tR = 1.63 min), the analysis took as long as 10 min, due to the positive element associated with column rinsing and conditioning. In subsection nine, the ultra-high-performance liquid chromatograph used was evaluated as energy efficient. The next step was to evaluate the use of chemicals from renewable sources; in the developed method, this criterion was met by water. According to the eleventh principle, toxic reagents should be replaced with more environmentally friendly alternatives or a reduced consumption. The developed method used ethanol as an organic modifier of the mobile phase. This is a non-toxic and biodegradable solvent. The last subsection considered operator safety and environmental risks, within which special attention should be paid to ethanol used in small amounts due to its flammable properties. The result for the environmental assessment of the developed method, according to the AGREE calculator, is listed in Figure 2. It contains the evaluation of both each parameters individually and the overall results of the whole procedure placed in the center of the pictogram—in the form of color and numerically. Ideally, the green method would have a value of 1; the value obtained was 0.71. The color is green, so the developed procedure fits into the principles of green analytical chemistry [30,31,32].

## 4. Conclusions

The developed method applied an efficient liquid chromatograph equipped with a short column with reduced particle size. It allowed for the obtainment of a good separation and reduced analysis time. The elimination of toxic organic solvents from the mobile phase during the chromatographic analysis and from the sample preparation step caused the proposed method to generate only biodegradable waste. The obtained method confirmed that caffeine may be determined using only environmentally friendly solvents, ethanol, and water.

The developed method was characterized by a very low limit of quantitation, equal to 0.047 µg/mL, and good reproducibility (a relative standard deviation lower than 1.1%). The obtained results show that, in most cases, the caffeine content in tested dietary supplements is 4–35% higher than the declared amount. In comparison, the caffeine content of the drug determined using this method was performed with an accuracy of 0.45%.

## Figures and Tables

**Figure 1 foods-12-02474-f001:**
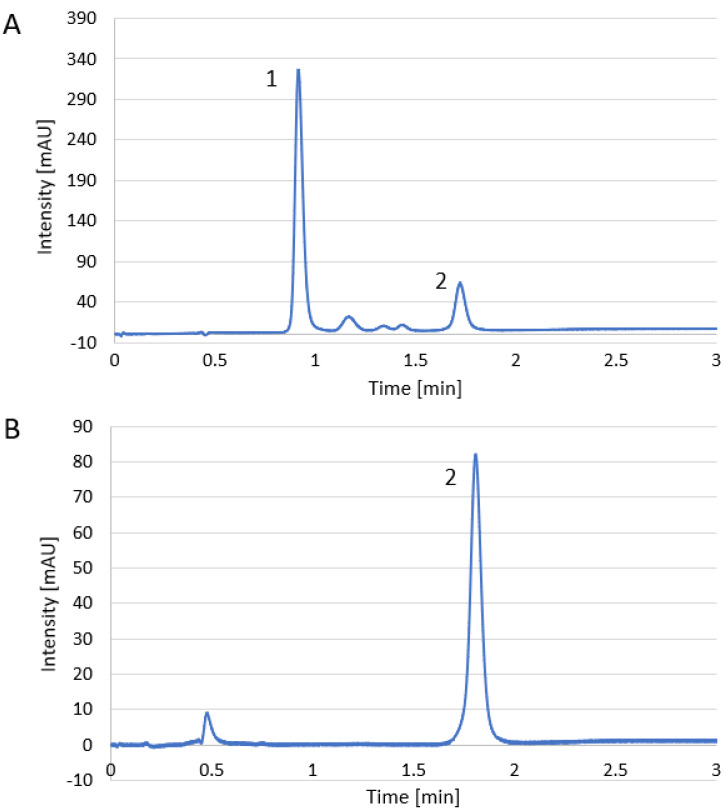
Exemplary chromatogram of R1 (**A**), and dietary supplement (**B**): 1-paracetamol and 2-caffeine.

**Figure 2 foods-12-02474-f002:**
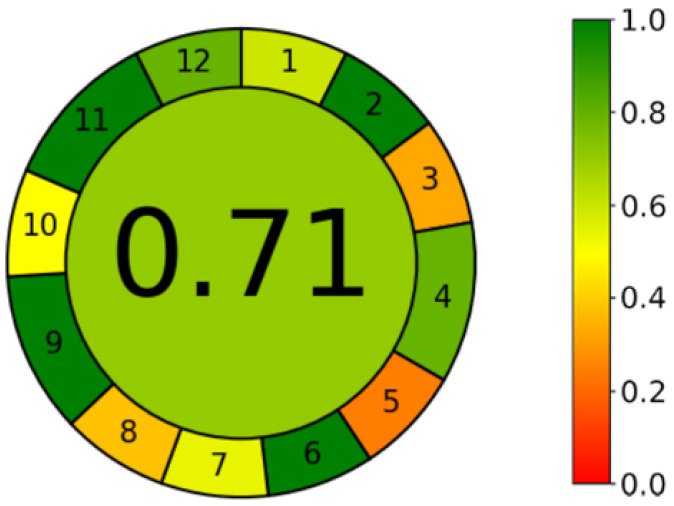
The result of the environmental assessment of the whole developed method according to AGREE. Colors represent environmental friendliness from 0 (red) to 1 (green).

**Table 1 foods-12-02474-t001:** Preparation form and ingredients of tested products.

	Preparation Form	Ingredients
R1	tablet	Active substances: paracetamol and caffeine. Excipients: povidone, potato starch, microcrystalline cellulose, sodium starch glycolate, talc, colloidal anhydrous silica, and magnesium stearate.
Dietary supplements		
#1	tablet	Anhydrous caffeine, calcium (as dicalcium phosphate), stearic acid, microcrystalline cellulose, croscarmellose sodium, magnesium stearate, silicon dioxide, and pharmaceutical glaze (shellac, povidone)
#2	capsule	Dextrose, anhydrous caffeine, capsule (bovine gelatin), and color (titanium dioxide)
#3	capsule	Maltodextrins, anhydrous caffeine, capsule shell (gelatin, dye-titanium dioxide), grapefruit extract, and anti-caking agent (magnesium salts of fatty acids)
#4	capsule	Anhydrous caffeine, bulking agent: cellulose, capsule shell (gelatin), anti-caking agents: magnesium salts of fatty acids and silicon dioxide
#5	capsule	caffeine, bulking agent (microcrystalline cellulose), anti-caking agent (magnesium salts of fatty acids), and shell (gelatin, dyes: cochineal red, sunset yellow FCF, quinoline yellow, and titanium dioxide)
#6	capsule	L-carnitine tartrate, citrus aurantium bitter orange fruit extract, cellulose capsule shell (hydroxypropyl methylcellulose dye: titanium dioxide), camellia sinensis green tea leaf extract, anhydrous caffeine, garcinia cambogia tamarind fruit extract, cayenne pepper capsicum annuum, n- acetyl l-tyrosine, and anti-caking agent: magnesium salts of fatty acids, black pepper fruit extract, piper nigrum, and chromium picolinate

**Table 2 foods-12-02474-t002:** Validation parameters of the developed method.

Parameter	Value
Calibration curve equation	y = 8184.40x + 7.72
Determination coefficient R2	0.9998
Linearity (mg/mL)	4.7 × 10^−5^ − 0.25
LOD (µg/mL)	0.016
LOQ (µg/mL)	0.047

**Table 3 foods-12-02474-t003:** Results of the accuracy test.

Average Mass of Tablet ± SD (mg)	Theoretical (Declared) Content in Tablet (mg)	Determined Mass of Tablet (mg)	Relative Standard Deviation (%)	Determined Caffeine in Tablet Versus Declared Value (%)
678.74 ± 2.35	50.0	50.23 ± 0.53	1.06	100.45

**Table 4 foods-12-02474-t004:** Intra-day and inter-day precision.

	1st Day	3rd Day	7th Day
	75 µL standard addition		
Concentration ± SD (µg/mL)	38.26 ± 0.06	38.55 ± 0.21	38.58 ± 0.11
RSD (%)	0.15	0.55	0.29
	150 µL standard addition		
Concentration ± SD (µg/mL)	49.16 ± 0.12	49.40 ± 0.11	49.43 ± 0.04
RSD (%)	0.24	0.22	0.08
	225 µL standard addition		
Concentration ± SD (µg/mL)	59.49 ± 0.14	59.34 ± 0.09	59.38 ± 0.14
RSD (%)	0.23	0.15	0.23

**Table 5 foods-12-02474-t005:** Results of dietary supplement analysis.

	The Average Mass of Tablet ± SD (mg)	Theoretical (Declared) Content in Tablet (mg)	Determined Mass of Tablet (mg)	Relative Standard Deviation (%)	Determined Caffeine in Tablet Versus Declared Value (%)
#1	671.65 ± 10.78	200.0	246.85 ± 2.89	1.17	123.43
#2	410.47 ± 22.52	100.0	119.40 ± 2.90	2.43	119.40
#3	589.94 ± 24.65	200.0	208.89 ± 2.40	1.15	104.44
#4	415.41 ± 5.51	200.0	263.74 ± 8.82	3.35	131.87
#5	388.47 ± 15.12	200.0	252.36 ± 4.10	1.63	126.18
#6	488.67 ± 5.66	66.7	90.77 ± 1.25	1.38	136.09

## Data Availability

The data presented in this study are available on request from the corresponding author.

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
