# Peer review of "A Determination of the Caffeine Content in Dietary Supplements According to Green Chemistry Principles"

_foods, 2023, doi:10.3390/foods12132474_

Round 1

Reviewer 1 Report

Review Report

Manuscript ID: foods-2464118

Title: Determination of caffeine content in dietary supplements according to green chemistry principles

The authors of this manuscript proposed a novel UHPLC method for the determination of caffeine content in dietary supplements. The presented methodology is in accordance with the rules of green analytical chemistry. The proposed method involves only water and ethanol for sample preparation and chromatographic separation with a short C18 column. Results have confirmed that caffeine can be quantitatively analyzed using these solvents. The method is characterized by the low limit of quantitation equal and good reproducibility.

 This study has a high potential to be cited. 

I recommend that the Editorial Office consider this manuscript after minor revision.

Reviewer’s Suggestion

In the section "3.3 Sample analysis" Can the authors give some statement or suggestion why "the determined amount of caffeine was higher than the declared value for all products."

In the section "3.4 Ecological aspects of the developed method". Statements in this section must be followed with relevant references. 

Author Response

Reviewer’s Suggestion

In the section "3.3 Sample analysis" Can the authors give some statement or suggestion why "the determined amount of caffeine was higher than the declared value for all products."

  • explanation of potential reason is provided in the manuscript.

In the section "3.4 Ecological aspects of the developed method". Statements in this section must be followed with relevant references. 

  • this part is a description of AGREE calculator. Description with literature is described in point 2.4.6. Some additional references are also provided.

Reviewer 2 Report

The manuscript describes the determination of caffeine using a green chemistry approach. The developed analytical method is simple and can be used in routine laboratories. The advantage of this method is its low sensitivity and the use of more environmentally friendly reagents.

The manuscript can be accepted after a minor revision. Please find specific comments underneath:

Lines 161-162: this sentence is confusing. How the inter-day precision was determined: using samples prepared on the day of analysis or re-injection of samples from the previous day(s)?

Table 3: units are missing for ‘Average mass of tablet’. It should be specified more clearly to what ‘Theoretical content’ and ‘Determined mass’ refer. The same remark applies to Table 5.

Table 4: what is the value of inter-day precision (in RSD,%)? For the moment only RSD’s for intra-day precision are shown.

Line 282: why is ethanol referred to as non-toxic?

Line 283: no safety risks were identified for the operator or the environment. Please specify how it was achieved as the work implied a certain exposure to ethanol.

Figure 2: the meaning of colours in the figure should be provided

Line 304: some values are marked in red

Author Response

Lines 161-162: this sentence is confusing. How the inter-day precision was determined: using samples prepared on the day of analysis or re-injection of samples from the previous day(s)?

  • It was a mistake, of course; it was a re-injection. It is corrected in the manuscript.

Table 3: units are missing for ‘Average mass of tablet’. It should be specified more clearly to what ‘Theoretical content’ and ‘Determined mass’ refer. The same remark applies to Table 5.

  • Columns caption in Table 5 were corrected.

Table 4: what is the value of inter-day precision (in RSD,%)? For the moment only RSD’s for intra-day precision are shown.

  • Additional data were provided.

Line 282: why is ethanol referred to as non-toxic?

  • we do not identify any toxic properties of ethanol in a water solution of 10-20%.

Line 283: no safety risks were identified for the operator or the environment. Please specify how it was achieved as the work implied a certain exposure to ethanol.

  • We use a water solution of ethanol with low concentration. We did not find any risk to the operator or the environment. However, if the Reviewer can identify it, we make the correction to our work. 

Figure 2: the meaning of colours in the figure should be provided

  • The explanation is provided in the figure caption.

Line 304: some values are marked in red

  • it was a mistake. It is corrected in the manuscript.

Reviewer 3 Report

In this paper, the authors focused on developing a method for the detection of caffeine in dietary supplements using UHPLC with UV detection. The main objective was to make the developed method compliant with the rules for green analytical chemistry. For this purpose, they tested two columns.  Fortis H2o column and a conventional C18 column for this purpose. For the gradient elution they chose ethanol and water as mobile phase. They validated the method and applied it to the determination of caffeine in six dietary supplements. The greenness of the method was done according to AGREE calculator and is discussed in the paper.

The work is well done and clearly written, but I have doubts about its suitability for this journal. The method developed does not bring any major novelty in terms of caffeine analysis. Rather, the contribution is more in terms of green analytical chemistry, so I would choose a journal with this focus.

Minor comments: 
fig. 1: the chromatograms are probably swapped, they do not match the description in the text or the picture description

fig 1: why didn't the authors use the original chromatogram and redraw it instead?

Quality of English languague is good.

Author Response

The work is well done and clearly written, but I have doubts about its suitability for this journal. The method developed does not bring any major novelty in terms of caffeine analysis. Rather, the contribution is more in terms of green analytical chemistry, so I would choose a journal with this focus.

  • the last point of Foods scope is "Food and environment" thus, we believe that our manuscript fulfills this point.

Minor comments: 
fig. 1: the chromatograms are probably swapped, they do not match the description in the text or the picture description

  • Thank you for your valuable insight. It was a mistake during editing. The manuscript is corrected. 

fig 1: why didn't the authors use the original chromatogram and redraw it instead?

  • we prefer to export data and plot it, as shown in the manuscript. If Reviewer does not agree with it we can change it.

Round 2

Reviewer 3 Report

 I am satisfied with the authors' answer.

English languague is ok.